# Response to Lenvatinib Is Associated with Optimal RelativeDose Intensity in Hepatocellular Carcinoma: Experience in Clinical Settings

**DOI:** 10.3390/cancers11111769

**Published:** 2019-11-10

**Authors:** Ryu Sasaki, Masanori Fukushima, Masafumi Haraguchi, Satoshi Miuma, Hisamitsu Miyaaki, Masaaki Hidaka, Susumu Eguchi, Satoshi Matsuo, Kazuaki Tajima, Toshihisa Matsuzaki, Satsuki Hashimoto, Kazuo Ooba, Yuki Kugiyama, Hiroshi Yatsuhashi, Yasuhide Motoyoshi, Masaya Shigeno, Noboru Kinoshita, Kazuhiko Nakao

**Affiliations:** 1Department of Gastroenterology and Hepatology, Nagasaki University Graduate School of Biomedical Sciences, 1-7-1 Sakamoto, Nagasaki City, Nagasaki 852-8501, Japan; ma-fukushima@nagasaki-u.ac.jp (M.F.); mharaguchi@nagasaki-u.ac.jp (M.H.); miuma1002@gmail.com (S.M.); miyaaki-hi@umin.ac.jp (H.M.); kazuhiko@nagasaki-u.ac.jp (K.N.); 2Department of Surgery, Nagasaki University Graduate School of Biomedical Sciences, 1-7-1 Sakamoto, Nagasaki City, Nagasaki 852-8501, Japan; mahidaka@nagasaki-u.ac.jp (M.H.); sueguchi@nagasaki-u.ac.jp (S.E.); 3Gastroenterology and Hepatology, Sasebo City General Hospital, 9-3 Hirase-cho, Sasebo City, Nagasaki 857-8511, Japan; s.matsuo41@gmail.com (S.M.); tajimax0527@gmail.com (K.T.); tmatsuzaki6@gmail.com (T.M.); 4Gastroenterology and Hepatology, Japan Community Health Care Organization, Isahaya General Hospital, 24-1 Eishohigashi-machi, Isahaya City, Nagasaki 854-8501, Japan; peach_chocolate510@yahoo.co.jp (S.H.); okazuo_ngs@yahoo.co.jp (K.O.); 5Clinical Research Center, National Hospital Organization Nagasaki Medical Center, 2-1001-1 Kubara, Oomura City, Nagasaki 856-8562, Japan; kugiyama.yuki.kn@mail.hosp.go.jp (Y.K.); yatsuhashi.hiroshi.wk@mail.hosp.go.jp (H.Y.); 6Gastroenterology and Hepatology, Nagasaki Harbor Medical Centor, 6-39 Shinchi-machi, Nagasaki City, Nagasaki 850-8798, Japan; motoyoshi-gi@umin.net; 7Gastroenterology and Hepatology, Japanese Red Cross Nagasaki Genbaku Hospital, 3-15 Mori-machi, Nagasaki City, Nagasaki 852-8511, Japan; mshi1010@nagasaki-med.jrc.or.jp; 8Gastroenterology and Hepatology, Sasebo Chuo Hospital, 15 Yamato-cho, Sasebo City, Nagasaki 857-1195, Japan; kinoshita@hakujyujikai.or.jp

**Keywords:** hepatocellular carcinoma, lenvatinib, relative dose intensity

## Abstract

Background: Lenvatinib is currently available as the first-line treatment for advanced unresectable hepatocellular carcinoma. We evaluated the relationship between its relative dose intensity (RDI) and response in clinical settings. Methods: From March 2018 to May 2019, 93 patients were administered lenvatinib at the Nagasaki University Hospital and its related facilities. Among these, 81 patients (66 men, 15 women, median age 72.0) who received lenvatinib were analyzed retrospectively. Results: Fourteen patients were Child–Pugh grade B, and 15 had received other systemic therapy. According to Response Evaluation Criteria in Solid Tumors (RECIST), the objective response (OR) rate was 17.3%. The overall survival (OS) was significantly better in the OR group (*p* = 0.011). There was a significant difference in RDI between the OR and non-OR groups (*p* < 0.05). The area under the receiver operating characteristics curve for OR prediction by the 4, 8, 12, and 16-week RDI were 0.666, 0.747, 0.731, and 0.704, respectively. In the 8-week RDI ≥67.0% group, OS was significantly better than in the 8-week RDI <67.0% group (*p* = 0.003). Conclusions: Because a sufficient RDI is required to achieve an OR, it is strongly recommended that lenvatinib should be administered to patients with good hepatic function and status.

## 1. Introduction

Hepatocellular carcinoma (HCC) is one of the most common causes of cancer-related deaths globally [1,2,3,4]. Systemic therapy, including molecular target agents, is the main treatment for advanced unresectable HCC [3,4]. In recent years, multiple drugs, such as regorafenib [5], lenvatinib [6], ramucirumab [7], and cabozantinib [8] in addition to sorafenib [9,10], have been shown to be effective for HCC. Of these molecular target agents, sorafenib and lenvatinib are currently available as the first-line treatment for advanced unresectable HCC [6,9,10]. The results of lenvatinib in clinical settings have been recently reported [11,12], and the importance of the relative dose intensity (RDI) has been noted [13]. However, the relationship between dosage and treatment effect in patients with HCC in clinical practice has not fully elucidated and remains unclear. The purpose of this retrospective study was to evaluate the relationship between the RDI and the patients’ response to lenvatinib in clinical settings.

## 2. Materials and Methods

### 2.1. Patient Characteristics

From March 2018 to May 2019, a total of 93 patients were evaluated who were administrated lenvatinib at the Nagasaki University Hospital and its related facilities. The exclusion criteria for this study were (1) a short observation period (<2 months, n = 6), (2) absence of proper image analysis (n = 2), and (3) insufficient archival material (n = 4). After the exclusion criteria were applied, data on 81 patients administrated lenvatinib were analyzed retrospectively. HCC was diagnosed based on its typical vascular patterns on either contrast-enhanced computed tomography (CT) or contrast-enhanced magnetic resonance imaging (MRI). Advanced HCC was defined as Barcelona Clinic liver cancer (BCLC) stage C plus stage A/B that was not amenable to curative treatment. Cases that were excluded by REFLECT trial [6], such as patients with ≥50% of their liver occupied by the tumor, obvious invasion of the bile duct, invasion of the main portal vein, and those who had received previous systemic therapy were included in the analysis.

### 2.2. Treatment Protocol and Relative Dose Intensity

Lenvatinib (Lenvima^®^; Eisai Co., Ltd., Tokyo, Japan) was orally administered to patients with unresectable HCC. The dose of lenvatinib was set based on body weight and hepatic reserve and was administered at an initial dosage of 12 mg/day for those over 60 kg and 8 mg/day for those under 60 kg. In the case of Child–Pugh grade B, the initial dose was 8 mg. Dose reduction was performed at the discretion of each facility. Lenvatinib continued until disease progression, unacceptable toxicity, or withdrawal of consent.

The RDI was calculated by dividing the actual dose by the ideal dose. The dose intensity was calculated for each week.

### 2.3. Evaluation Criteria for Adverse Events and Response

Dose interruption and sequential reduction of lenvatinib for drug-related adverse events (AEs) were performed according to the guidelines for the administration of lenvatinib. AEs were assessed using the National Cancer Institute Common Terminology Criteria for Adverse Events, version 4.0. 

Treatment response was evaluated by the use of a triphasic scanning technique, CT or MRI, in accordance with Response Evaluation Criteria in Solid Tumors (RECIST) [14] and modified RECIST (mRECIST) [15]. Tumors were evaluated once within the first 8 weeks and then every 8 weeks thereafter. The radiological response was defined as the best response based on the effect assessment after 8 weeks. The objective response (OR) was defined as complete response (CR) plus partial response (PR). Disease control (DC) was defined as OR plus stable disease (SD). The relationship between RDI and response was analyzed using RECIST, which is more objective than mRECIST.

### 2.4. Ethical Considerations

Informed consent to use medical records was obtained from each patient. These processes and the study protocol were approved by the Ethical Committee of our institution (confirmation number: 19041523-2) in accordance with the Declaration of Helsinki. Our research is available on the Clinical Research Center, Nagasaki University Hospital website (http://www.mh.nagasaki-u.ac.jp/research/rinsho/patients/open_gastro.html).

### 2.5. Statistical Analysis

Continuous variables were dichotomized with respect to the median value. The chi-squared and Fisher’s exact tests were used to compare categorical data. A comparison between groups for continuous variables was done by the Student’s *t*-test and the Mann–Whitney U test if necessary. The predictive value of the RDI to the radiological response was assessed by receiver operating characteristic curves and calculating the area under the receiver operator curve (AUROC). The optimal cutoff RDI value was determined based on the optimal sensitivity and specificity. Survival curves were assessed by the Kaplan–Meier method. The survival curves were compared using the log-rank test. A *p-*value of 0.05 was considered statistically significant. Data analysis was performed with SPSS version 22.0 (IBM Corp., Armonk, NY, USA).

## 3. Results

### 3.1. Patient Characteristics

The baseline characteristics of the 81 patients with HCC (66 men, 15 women) who were included in this study are summarized in Table 1. The median age of the patients was 72.0 years. Of the 81 patients, 14 patients were Child–Pugh grade B (17.2%), and 34 patients had extrahepatic spread (41.9%). A total of 14 patients had macroscopic portal vein invasion (17.2%), and 15 patients had undergone other systemic therapy (18.5%) and were not included in the REFLECT study. Regarding the previous systemic chemotherapy, all 15 patients were treated with sorafenib.

### 3.2. Adverse Events

In this study, 97.5% of patients had all-grade AEs, and 43.2% of patients had grade ≥3 AEs (Table 2). The incidence of all AE grades of hypertension, fatigue, decreased appetite, hypothyroidism, and proteinuria during the observation period was 61.7%, 58.0%, 56.7%, 51.8%, and 45.6%, respectively. In terms of grade ≥3 AEs, proteinuria and elevated liver enzymes occurred more frequently in our study than in the REFLECT study. A total of 68 patients (83.9%) had dose reduction or withdrawal owing to AEs.

### 3.3. Response

According to RECIST 1.1, 3 patients (3.7%) had CR, 11 patients had PR (13.6%), and 38 patients had SD (46.9%). The OR rate was 17.3%, and the DC rate was 64.2% (Table 3). Table 3 also presents the findings of mRECIST. The response, according to the Child–Pugh score and performance status (PS), are summarized in Table 4. The OR rate in patients who were Child–Pugh grade A was 20.8%, whereas it was 0.0% in patients who were Child–Pugh grade B. The OR rate in patients with a performance status of 0 was 25.0%, whereas it was 6.0% in patients with a performance status of 1/2.

### 3.4. Overall Survival

The median overall survival (OS) was 11.6 months. The OS was compared in the two groups stratified by radiological response. The OS was significantly better in the OR group (*p* = 0.011) (Figure 1). Further, the OS was divided into two groups stratified by the Child–Pugh score and performance status (Figure 2). The OS was significantly better in the Child–Pugh grade A group (*p* < 0.001) and in the PS 0 group (*p* < 0.001).

### 3.5. Relative Dose Intensity and Radiological Response

The RDI for each week is shown in Figure 3A. The RDI decreased as the number of weeks passed, and the overall rate was 61.1%. In Figure 3B, the RDI is divided into two groups: patients with and without OR. There was a significant difference in the RDI between the two groups from 8 weeks onwards. The median RDI at week 8 was 67.0%, and when it was divided into two groups with a median of 67.0%, the change in size from the baseline was as shown in Figure 4. 

### 3.6. Predictive Ability of Relative Dose Intensity for Radiological Response

We examined the predictive ability of RDI for a radiological response. Figure 5 shows the area under the ROC curve (AUROC) for OR prediction by RDI. The AUROC for OR prediction by the 4, 8, 12, and 16-week RDI were 0.666, 0.747, 0.731, and 0.704, respectively. The AUROC increased until 8 weeks and then plateaued. Therefore, we examined whether 8-week RDI is useful for predicting OR achievement. The cutoff values are those giving the highest sum of sensitivity plus specificity. The optimal cutoff values for RDI were 90% (sensitivity 77.6%, specificity 74.4%) to achieve OR.

### 3.7. Overall Survival Stratified by Relative Dose Intensity

OS was divided into two groups according to an RDI of 67.0%, which was the median value of the 8-week RDI (Figure 6). In the 8-week RDI ≥67.0% group, OS was significantly better than in the 8-week RDI< 67.0% group (*p* = 0.003). Table 5 compares the high and low 8-week RDI groups. There was a significant difference in BMI, PS, Child–Pugh grade, BCLC stage, platelet count, prothrombin time, albumin, and dose reduction at administration between the low and high 8-week RDI groups. Child–Pugh grade B and poor PS cases tended to have low RDI (Figure 7).

## 4. Discussion

Lenvatinib is a tyrosine kinase inhibitor that targets VEGF receptors 1–3, fibroblast growth factor receptors 1–4, PDGFRα, RET, and KIT [16,17]. The phase 3 clinical trial, REFLECT [6], was the first study to show that lenvatinib was non-inferior to sorafenib for OS in patients with advanced HCC. Unlike other cancer types, lenvatinib pharmacokinetics in patients with HCC were affected by body weight, supporting a body weight-based dosing approach [18,19].

In the treatment of thyroid cancer with lenvatinib, control of potential AEs and dose optimization are important [13]. In clinical practice, lenvatinib for HCC treatment is administered to more patients with unfavorable conditions, such as those with poor hepatic reserve or poor PS, than in the REFLECT study. Hence, the lenvatinib dose used often differs from the recommended dose. It is necessary to verify how much lenvatinib is required to achieve efficacy.

Our study showed that the required dose was significantly higher in OR cases. The dose after the first 8 weeks was especially important (Figure 3B). We also examined which week's RDI was most useful for predicting the achievement of OR. The AUROC for predicting OR by RDI almost plateaued at 8 weeks (Figure 5). The results suggest that 8-week RDI is an important evaluation point. We demonstrated that a high anti-tumor effect can be expected if the 8-week RDI is sufficient. In addition, if the 8-week RDI is sufficient, an extension of OS can be expected owing to a high anti-tumor effect.

Tyrosine kinase inhibitor dosage and therapeutic effects have been reported in other cancers [20,21]. In the treatment of HCC, a relationship between regorafenib and RDI has been reported [22]. In addition, Takahashi, et al. reported the RDI of lenvatinib in HCC treatment [23]. Their report is consistent with our finding that sufficient RDI is important for therapeutic efficacy. Our results are significant in that they provide evidence that the evaluation period should be 8 weeks (Figure 3; Figure 5), involve a more objective size reduction according to the RECIST criteria (Figure 4), and demonstrate a significant difference in OS stratified by RDI (Figure 6).

In our study, the OR rate of 17.3% and DC rate of 64.2%, according to RECIST, were consistent with the results from the REFLECT trial. In addition, the therapeutic effect varied according to the hepatic reserve. In particular, patients with Child–Pugh grade B did not achieve OR, and their OS was poor. Moreover, even in cases with poor PS, the OR rate, as well as the OS, was low. Child–Pugh grade B and poor PS cases tended to have low RDI (Figure 7). Our results suggest that a low RDI was presumed to affect the therapeutic effect. Therefore, to maintain a sufficient RDI, it is essential to administer lenvatinib to patients with Child–Pugh grade A and PS 0. 

One of the limitations of the present study was its retrospective nature. Additionally, in our study, the relationship between DC and RDI was weak. It is unclear whether continuing at a low RDI will contribute to SD and an improved prognosis. A further increase in the number of cases and an extension of the observation period is necessary. In addition, combination therapy involving lenvatinib and other treatments should be considered in the future. Regardless of these limitations, our data are thus the first to demonstrate the relationship between RDI and anti-tumor effects according to the RECIST criteria in a clinical setting.

## 5. Conclusions

In conclusion, our study revealed that a sufficient RDI is required to achieve OR and that OR extends prognosis. It is strongly recommended that lenvatinib be administered at a sufficient dose for patients in good condition.

## Figures and Tables

**Figure 1 cancers-11-01769-f001:**
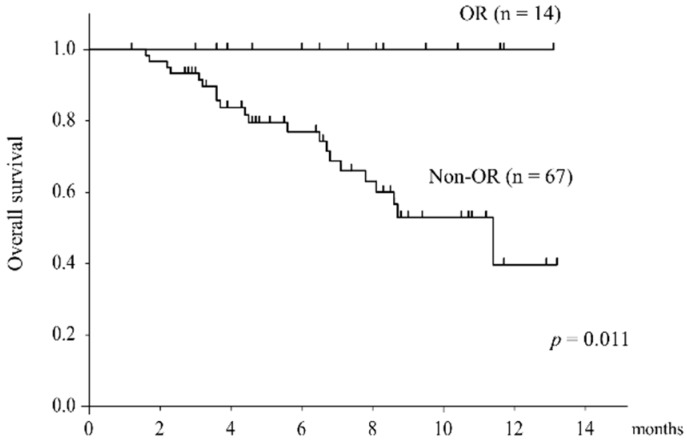
Overall survival stratified by radiological response according to Response Evaluation Criteria in Solid Tumors (RECIST). Kaplan–Meier curves for OR (objective response) and non-OR. There was a significant difference in OS between the two groups (*p* = 0.011).

**Figure 2 cancers-11-01769-f002:**
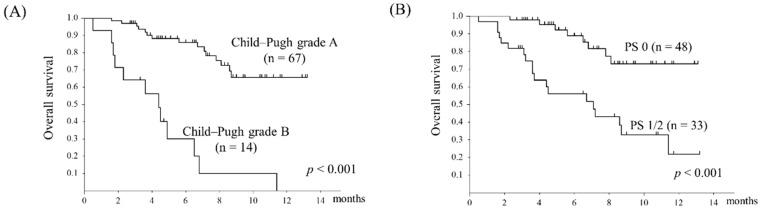
Overall survival stratified by (**A**) Child–Pugh grade and (**B**) performance status. (**A**) Kaplan–Meier curves for Child–Pugh grade A and Child–Pugh grade B. There was a significant difference in OS between the two groups (*p* < 0.001). (**B**) Kaplan–Meier curves for performance status 0 and performance status 1/2. There was a significant difference in OS between the two groups (*p* < 0.001).

**Figure 3 cancers-11-01769-f003:**
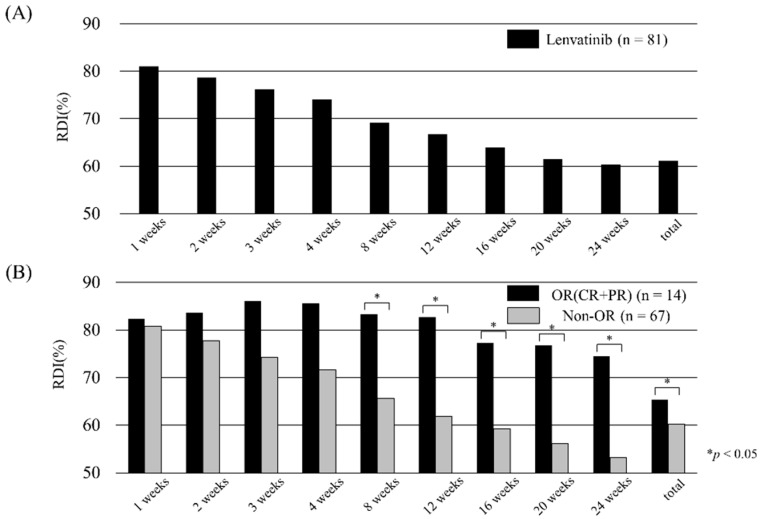
Relative dose intensity (RDI) of lenvatinib (**A**) for all patients and (**B**) stratified by radiological response. (A) RDI for each week. The overall RDI was 61.1%. (B) RDI for each week stratified by objective response (OR; black bars) and non-OR (gray bars). After 8 weeks, the OR group tended to have a significantly higher RDI (*p* < 0.05).

**Figure 4 cancers-11-01769-f004:**
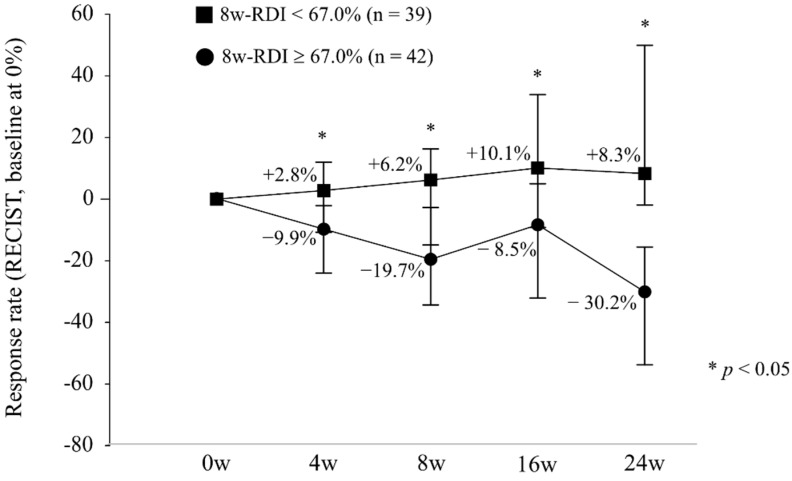
Response rate stratified by relative dose intensity at 8 weeks. In the 8-week RDI< 67.0% subgroup (black square), the response rate was +2.8%, +6.2%, +10.1%, and +8.3% at 4, 8, 16, and 24 weeks, respectively. In the 8-week RDI ≥ 67.0% subgroup (black circle), the response rate was −9.9%, −19.7%, −8.5%, and −30.2% at 4, 8, 16, and 24 weeks, respectively. The response rate was significantly different each week (*p* < 0.05).

**Figure 5 cancers-11-01769-f005:**
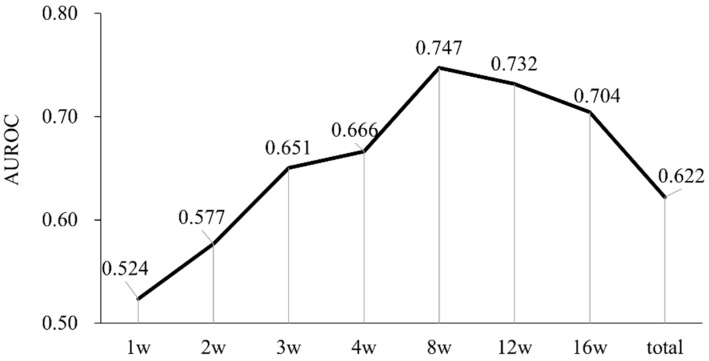
Discrimination ability of relative dose intensity for radiological response by the area under the receiver operating curve. Plot of the area under the receiver operator curve (AUROC) identifying radiological response by RDI for each week. Maximum AUROC was 0.747 at 8 weeks.

**Figure 6 cancers-11-01769-f006:**
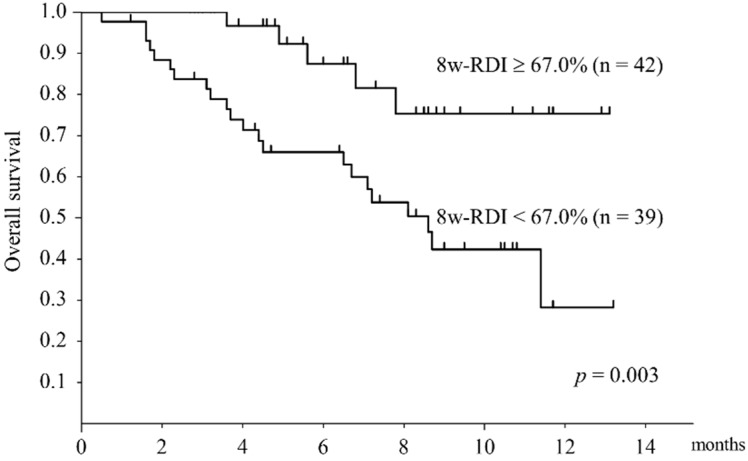
Overall survival stratified by relative dose intensity at 8 weeks. Kaplan–Meier curves stratified by an 8-week RDI of 67.0%. There was a significant difference in overall survival (OS) between the two groups.

**Figure 7 cancers-11-01769-f007:**
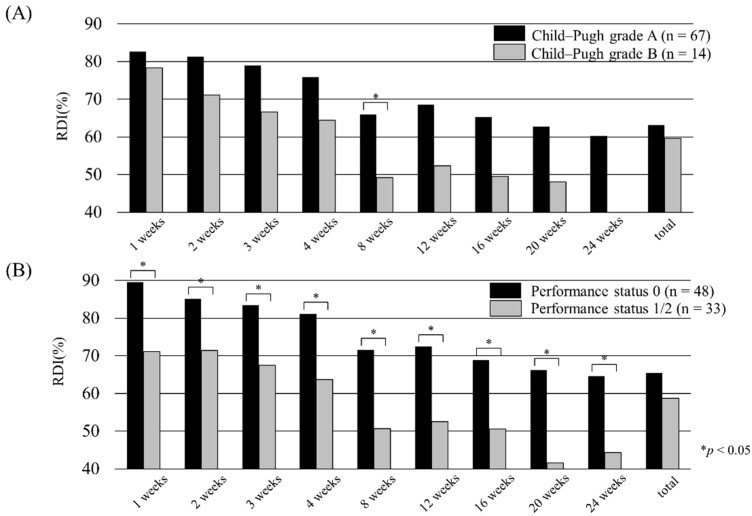
Relative dose intensity of lenvatinib stratified by (**A**) Child–Pugh grade and (**B**) radiological response. (**A**) RDI for each week stratified by Child–Pugh grade A (black bars) and Child–Pugh grade B (gray bars). The Child–Pugh grade A group tended to have significantly higher RDI at 8 weeks (*p* < 0.05). In the Child–Pugh grade B group, the number of cases at 24 weeks was not available and could not be calculated. (B) RDI for each week stratified by performance status (PS) 0 (black bars) and PS 1/2 (gray bars). The PS 0 group tended to have a significantly higher RDI (*p* < 0.05).

**Table 1 cancers-11-01769-t001:** Characteristics of the patients enrolled in the present study.

Variable		n = 81
Age	year	72.0 (41–88)
Gender	male/female	66/15
BMI	kg/m^2^	24.10 (17.5–34.4)
Performance status	0/1/2	48/28/5
Child–Pugh grade	A/B	67/14
ALBI grade	1/2/3	20/57/4
Macroscopic PV invasion	Vp3 or Vp4	14 (17.2)
Extrahepatic spread	+	34 (41.9)
BCLC stage	A/B/C	1/19/61
Etiology	HBV/HCV/NBNC	22/26/33
Platelet count	×10^4^/μL	12.30 (4.0–38.1)
PT	%	88.0 (47–128)
T.bil	mg/dL	0.90 (0.4–2.6)
Albumin	g/dL	3.50 (2.2–4.6)
ALT	IU/mL	27.0 (11–198)
AFP	ng/ml	68.8 (1.8–23,124)
DCP	mAU/ml	484.0 (10–990,474)
Systemic therapy	naïve/experienced	66/15
Dose reduction	at administration	32 (39.5)

Data are given as the medians with ranges or numbers with percentages. BMI, body mass index; ALBI, Albumin bilirubin index; BCLC, Barcelona Clinic liver cancer; PV, portal vein; PT, prothrombin; T.bil, total bilirubin; ALT, alanine aminotransferase; AFP, alpha fetoprotein; DCP, Des-gamma-carboxy prothrombin.

**Table 2 cancers-11-01769-t002:** Frequency of treatment-emergent adverse events.

Treatment-Emergent Adverse Events	Any Grade	Grade ≥ 3
All adverse events	79 (97.5%)	35 (43.2%)
Hypertension	50 (61.7%)	11 (13.6%)
Fatigue	47 (58.0%)	1 (1.2%)
Decreased appetite	46 (56.7%)	3 (3.7%)
Hypothyroidism	42 (51.8%)	0 (0.0%)
Proteinuria	37 (45.6%)	7 (8.6%)
Palmar–plantar erythrodysesthesia syndrome	31 (38.2%)	0 (0.0%)
Thrombocytopenia	30 (37.0%)	3 (3.7%)
Elevated liver enzymes	25 (30.8%)	6 (7.4%)
Diarrhea	20 (24.6%)	0 (0.0%)
Weight loss	18 (22.2%)	1 (1.2%)
Dysphonia	12 (14.8%)	0 (0.0%)
edema	11 (13.5%)	1 (1.2%)
Rash	5 (6.1%)	0 (0.0%)
Loss of hair	1 (1.2%)	0 (0.0%)

**Table 3 cancers-11-01769-t003:** Summary of efficacy measures, according to modified Response Evaluation Criteria in Solid Tumors (mRECIST) and Response Evaluation Criteria in Solid Tumors (RECIST).

Response Category	Lenvatinib (n = 81)	
	mRECIST	RECIST
Complete response (CR)	7 (8.6%)	3(3.7%)
Partial response (PR)	21 (25.9%)	11 (13.6%)
Stable disease (SD)	23 (28.4%)	38 (46.9%)
Progressive disease (PD)	20 (24.7%)	23 (28.4%)
Unknown or not evaluable	10 (12.3%)	6 (7.4%)
Objective response (OR)	28 (34.6%)	14 (17.3%)
Disease control (DC)	51 (63.0%)	52 (64.2%)

**Table 4 cancers-11-01769-t004:** Summary of efficacy measures according to RECIST stratified by Child–Pugh grade and performance status.

Factors	Child–Pugh Grade		Performance Status	
	grade A (n = 67)	grade B (n = 14)	PS 0 (n =48)	PS 1/2 (n = 33)
Complete response (CR)	3 (4.4%)	0 (0.0%)	2 (4.2%)	1 (3.0%)
Partial response (PR)	11 (16.4%)	0 (0.0%)	10 (20.8%)	1 (3.0%)
Stable disease (SD)	32 (47.7%)	6 (42.9%)	24 (50.0%)	14 (42.4%)
Progressive disease (PD)	18 (26.8%)	5 (35.7%)	10 (20.8%)	13 (39.3%)
Unknown or not evaluable	3 (4.4%)	3 (21.4%)	1 (2.1%)	5 (15.1%)
Objective response (OR)	14 (20.8%)	0 (0.0%)	12 (25.0%)	2 (6.0%)
Disease control (DC)	46 (68.6%)	6 (42.9%)	36 (75.0%)	16 (48.4%)

**Table 5 cancers-11-01769-t005:** Comparison between two groups with relative dose intensity.

Variable	8w-RDI ≥ 67.0%	8w-RDI < 67.0%	*p*-Value
Age	year	71.0 (46–84)	76.0 (41–88)	0.065
Gender	male/female	37/5	29/10	0.111
BMI	kg/m^2^	24.65 (18.6–33.6)	22.15 (17.5–34.4)	0.019
Performance status	0/1/2	34/8/0	14/20/5	<0.001
Child–Pugh grade	A/B	38/4	29/10	0.055
ALBI grade	1/2/3	11/31/0	9/26/4	0.103
Macroscopic PV invasion	Vp3 or Vp4	7 (16.7)	7 (17.9)	0.878
Extrahepatic spread	+	16 (38.1)	18 (46.2)	0.462
BCLC stage	A/B/C	1/16/25	0/3/36	0.002
Etiology	HBV/HCV/NBNC	13/12/17	9/14/16	0.669
Platelet count	×10^4^/μL	14.85 (4.6–38.1)	10.60 (4.0–28.0)	0.015
PT	%	90.5 (64–128)	84.0 (47–118)	0.013
T.bil	mg/dL	0.90 (0.4–1.8)	1.00 (0.4–2.6)	0.307
Albumin	g/dL	3.70 (2.8–4.6)	3.50 (2.2–4.5)	0.028
ALT	IU/mL	27.5 (11–143)	26.0 (11–198)	0.909
AFP	ng/mL	43.0 (1.8–9926)	140.0 (1.9–23,124)	0.061
DCP	mAU/mL	572.5 (10–78884)	458.0 (11–990,474)	0.609
Systemic therapy	naïve/experienced	36/6	30/9	0.308
Dose reduction	at administration	9 (21.4)	23 (59.0)	<0.001
Results given as median (range) or n (%).

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
