# Peer review of "Response to Lenvatinib Is Associated with Optimal RelativeDose Intensity in Hepatocellular Carcinoma: Experience in Clinical Settings"

_cancers, 2019, doi:10.3390/cancers11111769_

Round 1

Reviewer 1 Report

The article "Relative dose intensity and response in real-world treatment with lenvatinib for advanced hepatocellular carcinoma" by Sasaki et al. studies lenvatinib in clinical settings and evaluates relationship between its relative dose and patient's response.

The paper is well written with results that will be of significant interest to the readers, especially from a clinical perspective. I do, however, have a few suggestions for the authors.

The use of the world "real-world treatment" in the title itself seems a bit unscientific; I would suggest a change in terminology. Could be replaced with "patients" or "clinics". I would also suggest expanding discussion part of the paper, for instance: apart from RECIST, what other similar studies have been done with other drugs (e.g. sorafenib) and maybe also in other cancer types? what were the results? how does it compare with the findings made by the authors?  I think the readers will also be interested in knowing what systemic therapies did the 15 patients (18.5%) receive. Do the authors propose a combination of lenvatinib with other therapeutic modalities?

Author Response

Comment 1

The article "Relative dose intensity and response in real-world treatment with lenvatinib for advanced hepatocellular carcinoma" by Sasaki et al. studies lenvatinib in clinical settings and evaluates relationship between its relative dose and patient's response. The paper is well written with results that will be of significant interest to the readers, especially from a clinical perspective. I do, however, have a few suggestions for the authors.

The use of the world "real-world treatment" in the title itself seems a bit unscientific; I would suggest a change in terminology. Could be replaced with "patients" or "clinics".

Response:

We thank the reviewer for this comment. As the reviewer noted, it would be more convincing to use “clinics” or “clinical setting”. We have changed the title from (p 1, line 2):

“Relative dose intensity and response in real-world treatment with lenvatinib for advanced hepatocellular carcinoma”

to

“Response to lenvatinib is associated with optimal relative dose intensity in hepatocellular carcinoma: Experience in clinical settings”

We have also made the following changes on (p 1, line 30):

“We evaluated the relationship between its relative dose intensity (RDI) and response in real-world treatment.”

to

“We evaluated the relationship between its relative dose intensity (RDI) and response in clinical settings.”

We have also changed title from (p 2, line 52):

“Recently, the results of lenvatinib in real-world data have also been reported [11,12]. The importance of the relative dose intensity (RDI) has been noted for lenvatinib [13].”

to

“The results of lenvatinib in clinical settings have been recently reported [11,12], and importance of the relative dose intensity (RDI) has been noted [13].”

We have also changed title from (p 2, line 56):

“The purpose of this retrospective study was to evaluate the relationship between the RDI and the patients’ response to lenvatinib in real-world treatment.”

to

“The purpose of this retrospective study was to evaluate the relationship between the RDI and the patients’ response to lenvatinib in clinical settings.”

We have also changed title from (p 2, line 69):

“This study was a real-world study, and dose reduction was performed at the discretion of each facility.”

to

“Dose reduction was performed at the discretion of each facility.”

We have also changed title from (p 10, line 230):

“Regardless of these limitations, our data are thus the first to demonstrate the relationship between RDI and anti-tumor effects in a real-world setting.”

to

“Regardless of these limitations, our data are thus the first to demonstrate the relationship between RDI and anti-tumor effects according to the RECIST criteria in a clinical setting.”

Comment 2

I would also suggest expanding discussion part of the paper, for instance: apart from RECIST, what other similar studies have been done with other drugs (e.g. sorafenib) and maybe also in other cancer types? what were the results? how does it compare with the findings made by the authors?

Response:

We wish to thank the reviewer for this comment. We have added the following text from on (p9, line 219):

“Tyrosine kinase inhibitor dosage and therapeutic effects have been reported in other cancers [20,21]. In the treatment of HCC, a relationship between regorafenib and RDI has been reported [22]. In addition, Takahashi el al. reported the RDI of lenvatinib in HCC treatment [23]. Their report is consistent with our finding that sufficient RDI is important for therapeutic efficacy. Our results are significant in that they provide evidence that the evaluation period should be 8 weeks (Figure 3, 5), involve a more objective size reduction according to the RECIST criteria (Figure 4), and demonstrate a significant difference in OS stratified by RDI (Figure 6).”

We have also changed the following text (p 2, line 54):

“However, the relationship between dosage and treatment effect in patients with HCC in clinical practice has not been reported.”

to

“However, the relationship between dosage and treatment effect in patients with HCC in clinical practice has not been fully elucidated and remains unclear.”

We have also added following references (p 11, line 298):

“20         Hayato S, Shumaker R, Ferry J, Binder T, Dutcus CE, Hussein Z: Exposure-response analysis and simulation of lenvatinib safety and efficacy in patients with radioiodine-refractory differentiated thyroid cancer. Cancer Chemother Pharmacol 2018;82:971-978.

21           Houk BE, Bello CL, Poland B, Rosen LS, Demetri GD, Motzer RJ: Relationship between exposure to sunitinib and efficacy and tolerability endpoints in patients with cancer: results of a pharmacokinetic/pharmacodynamic meta-analysis. Cancer Chemother Pharmacol 2010;66:357-371.

22           Wang W, Tsuchiya K, Kurosaki M, Yasui Y, Inada K, Kirino S, Yamashita K, Sekiguchi S, Hayakawa Y, Osawa L, et al.: Sorafenib-Regorafenib Sequential Therapy in Japanese Patients with Unresectable Hepatocellular Carcinoma-Relative Dose Intensity and Post-Regorafenib Therapies in Real World Practice. Cancers (Basel) 2019;11

23           Takahashi A, Moriguchi M, Seko Y, Ishikawa H, Yo T, Kimura H, Fujii H, Shima T, Mitsumoto Y, Ishiba H, et al.: Impact of Relative Dose Intensity of Early-phase Lenvatinib Treatment on Therapeutic Response in Hepatocellular Carcinoma. Anticancer Res 2019;39:5149-5156.20”

Comment 3

I think the readers will also be interested in knowing what systemic therapies did the 15 patients (18.5%) receive.

Response:

We wish to thank the reviewer for this comment. We agree that additional information about systemic therapies in those 15 patients would be valuable. We have added the following text to (p 3, line 108):

“Regarding the previous systemic chemotherapy, all 15 patients were treated with sorafenib.”

Comment 4

Do the authors propose a combination of lenvatinib with other therapeutic modalities?

Response:

We thank the reviewer for this insightful comment. The effects of combination treatment compared with lenvatinib monotherapy have not been previously reported. We are now investigating this point and intend to report it in the future. As proved by the TACTICS trial (Sorafenib+TACE), sorafenib in combination with TACE showed positive results. Our center is currently involved in a multicenter trial evaluating Lenvatinib and TACE. However, since this study did not involve combination therapy, I would like to mention it as in issue that should be considered in the future. We have added the following text on (p 10, line 230):

“In addition, combination therapy involving lenvatinib and other treatments should be considered in the future.”

Reviewer 2 Report

Relative dose intensity and response in real-world treatment with lenvatinib for advanced hepatocellular carcinoma (cancers-629481)

This is a retrospective study coming from Japan focused on the role of lenvatinib for the treatment of 81 patients with advanced hepatocellular carcinoma (HCC).

The study is of great interest, although the numerosity is relatively small. The results are extremely interesting.

Nevertheless, I have some issues that should be solved before the acceptance of the article.

Did the Authors use the inclusion criteria of the REFLECT study, or did they used enlarged criteria? I think a more detailed definition of the selection criteria should be reported in Material and Methods. In addition, a clear definition of advanced HCC should be clearly stated.

Why the Authors decided to evaluate the objective response according to the findings of the RECIST 1.1 instead of the more recently introduced mRECIST? I think the Authors should motivate this decision in Materials and Methods.

Figures 3 and 7 must be improved in their quality. I think they should be clearer if different plain colors are used instead of the actually used graphical solution.

Discussion is very short! I think more argumentation should be reported on the already demonstrated efficacy of lenvatinib for the treatment of advanced HCC. The existing literature reporting the connection between other anti-HCC drugs and relative dose intensity should be reported and discussed.

Author Response

Comment 1

Did the Authors use the inclusion criteria of the REFLECT study, or did they used enlarged criteria? I think a more detailed definition of the selection criteria should be reported in Material and Methods. In addition, a clear definition of advanced HCC should be clearly stated.

Response:

We thank the reviewer for this comment. As the reviewer noted, our original manuscript’s description of the REFLECT and inclusion criteria was insufficient. Furthermore, a definition of advanced HCC was not provided. We have added the following text on (p 2, line 65):

“HCC was diagnosed based on its typical vascular patterns on either contrast-enhanced computed tomography (CT) or contrast-enhanced magnetic resonance imaging (MRI). Advanced HCC was defined as BCLC stage C plus stage A/B that was not amenable to curative treatment. Cases that were excluded by REFLECT [6], such as patients with ≥50% of their liver occupied by the tumor, obvious invasion of the bile duct, invasion of the main portal vein, and those that had received previous systemic therapy were included in the analysis.”

Comment 2

Why the Authors decided to evaluate the objective response according to the findings of the RECIST 1.1 instead of the more recently introduced mRECIST? I think the Authors should motivate this decision in Materials and Methods.

Response:

We thank the reviewer for this pertinent comment. In our study, both RECIST and mRECIST were evaluated. However, in mRECIST, the definition of the stained area may vary greatly depending on the evaluator, and subjective factors may be introduced. Moreover, in the REFLECT study, the mRECIST rating was different between the investigator’s review and the masked independent imaging reviewer (OR 24% vs. 40%, while no significant difference was observed for DCR: 75.5% vs. 73.8%). In this report, the use of mRECIST itself is described as a limitation (Lancet 2018;391:1163-1173.). In our study, we wanted to simply verify the antitumor effect; therefore, we adopted RECIST, which guarantees objectivity. We have added the following text on (p 2, line 84 to clarify this):

“The relationship between RDI and response was analyzed using RECIST, which is more objective than mRECIST.”

Comment 3

Figures 3 and 7 must be improved in their quality. I think they should be clearer if different plain colors are used instead of the actually used graphical solution.

Response:

We wish to thank the reviewer for this comment. In accordance with their suggestion, we have changed Figure 3 to a plain color scheme as shown below:

 Attached please find the Figures.

We have also changed the following text on (p 6, line 157):

“(B) RDI for each week stratified by OR (lattice bars) and non-OR (dotted bars).”

to

“(B) RDI for each week stratified by objective response (OR; black bars) and non-OR (gray bars)”

We have changed Figure 7 as shown below:

We have also changed the following text on (p 9, line 194):

“(A) RDI for each week stratified by Child-Pugh grade A (lattice bars) and Child-Pugh grade B (dotted bars).”

to

“(A) RDI for each week stratified by Child-Pugh grade A (black bars) and Child-Pugh grade B (gray bars).”

And (p 9, line 197):

“(B) RDI for each week stratified by PS 0 (lattice bars) and PS 1/2 (dotted bars).”

to

“(B) RDI for each week stratified by PS 0 (black bars) and PS 1/2 (gray bars).”

Comment 4

Discussion is very short! I think more argumentation should be reported on the already demonstrated efficacy of lenvatinib for the treatment of advanced HCC. The existing literature reporting the connection between other anti-HCC drugs and relative dose intensity should be reported and discussed.

Response:

We wish to thank the reviewer for this comment. To address the current deficiencies in the discussion, we have added the following text (p 9, line 219):

“Tyrosine kinase inhibitor dosage and therapeutic effects have been reported in other cancers [20,21]. In the treatment of HCC, a relationship between regorafenib and RDI has been reported [22]. In addition, Takahashi el al. reported the RDI of lenvatinib in HCC treatment [23]. Their report is consistent with our finding that sufficient RDI is important for therapeutic efficacy. Our results are significant in that they provide evidence that the evaluation period should be 8 weeks (Figure 3, 5), involve a more objective size reduction according to the RECIST criteria (Figure 4), and demonstrate a significant difference in OS stratified by RDI (Figure 6),”

We have also changed following text on p 2, line 54:

“However, the relationship between dosage and treatment effect in patients with HCC in clinical practice has not been reported.”

to

“However, the relationship between dosage and treatment effect in patients with HCC in clinical practice has not been fully elucidated and remains unclear.”

--------------------------------------------

Additional modifications

--------------------------------------------

1)We have changed following text on (p 1, lin32):

“a total of 93 patients were administrated lenvatinib at Nagasaki University Hospital and its related facilities.”

to

“93 patients were administered lenvatinib at Nagasaki University Hospital and its related facilities.”

2)We have changed following text on (p 1, line 34):

“Fourteen patients were Child-Pugh grade B and 15 patients had undergone other systemic therapy.”

to

“Fourteen patients were Child-Pugh grade B and 15 had received other systemic therapy.

3)We have changed following text on (p 1, line 36)

“The OS was significantly better in the OR group (p = 0.011).”

to

“The overall survival (OS) was significantly better in the OR group (p = 0.011).

4)We have added following text (p 3, line 111):

“BMI, body mass index; ALBI, Albumin bilirubin index; BCLC, Barcelona Clinic liver cancer; PV, portal vein; PT, prothrombin; T.bil, total bilirubin; ALT, alanine aminotransferase; AFP, alpha fetoprotein; DCP, Des-gamma-carboxy prothrombin;”

5)We have changed following text on (p 9, line 202):

“The phase 3 clinical trial, REFLECT [6], was the first study that showed that lenvatinib was non-inferior to sorafenib for OS in patients with advanced HCC.

“The phase 3 clinical trial, REFLECT [6], was the first study to show that lenvatinib was non-inferior to sorafenib for OS in patients with advanced HCC.”

6)We have changed following text on (p 9, line 205):

“supporting body weight-based dosing [18,19].”

to

“supporting a body weight-based dosing approach [18,19].”

7)We have changed following text on (p 9, line 214):

“The AUROC predicting OR with RDI”

to

“The AUROC for predicting OR by RDI”

8)We have changed following text on (p 9, line 220):

“the therapeutic effect was different in hepatic reserve.”

“the therapeutic effect varied according to hepatic reserve.”

9)We have changed following text on (p 9, line 222):

“Moreover, even in cases with poor PS, the OR rate was low, and the OS was also poor.”

“Moreover, even in cases with poor PS, the OR rate, as well as the OS, was low”

10)We have changed following text on (p 10, line 227):

“Additionally, in our study, the relationship between DC and RDI was low.”

to

“Additionally, in our study, the relationship between DC and RDI was weak.”

11)We have changed following text on (p 10, line 234):

“In conclusion, our study revealed a sufficient RDI is required to achieve OR and that OR extends prognosis.”

“In conclusion, our study revealed that a sufficient RDI is required to achieve OR and that OR extends prognosis.”

All additions and modifications are indicated in red font, and removed passages are shown by strike-through.

We sincerely appreciate the academic editor and reviewers for their insightful comments on our paper. The revised manuscript is significantly improved over the initial submission. We hope that you find the revised manuscript suitable for publication.

Round 2

Reviewer 2 Report

The Authors improved the quality of the study. The paper can be accepted for publication. No other comments